# Zooplankton responses to simulated marine heatwave in the Mediterranean Sea using *in situ* mesocosms

**Soultana Zervoudaki**[1]*, **Maria Protopapa**[1], **Andriana Koutsandrea**[1,2], **Anna Jansson**[2], **Ella von Weissenberg**[3], **Georgios Fyttis**[4], **Athanasia Sakavara**[1], **Kostas Kavakakis**[1], **Charitomeni Chariati**[1], **Katja Anttila**[5], **Pauline Bourdin**[5], **Behzad Mostajir**[6], **Francesca Vidussi**[6], **Jonna Engström-Öst**[2]

**1** Hellenic Centre for Marine Research, Institute of Oceanography, Athens, Greece, **2** Novia University of Applied Sciences, Ekenäs, Finland, **3** Biocenter Finland, Faculty of Biological and Environmental Sciences, University of Helsinki, Helsinki, Finland, **4** Department of Biological Sciences, University of Cyprus, Nicosia, Cyprus, **5** Department of Biology, University of Turku, Turku, Finland, **6** Marine Biodiversity, Exploitation and Conservation, Ifremer, IRD, Université de Montpellier, Montpellier Cedex 05, France

* tanya@hcmr.gr

**Data Availability Statement:** Data are available at https://doi.org/10.5061/dryad.vt4b8gv1s

**Funding:** The work was funded by the AQUACOSM-plus project, which has received

## Abstract

Globally, marine heatwave frequency, intensity, and duration are on the rise, posing a significant threat to plankton communities, the foundational elements of the marine food web. This study investigates the ecological and physiological responses of a temperate plankton community in the Thau lagoon, north-western Mediterranean, to a simulated +3˚C ten-day heatwave followed by a ten-day post-heatwave period in *in-situ* mesocosms. Our analyses encompassed zooplankton grazing, production, community composition in water and sediment traps, as well as oxidative stress and anti-oxidant biomarkers. The results revealed increased abundances of harpacticoid copepods and polychaete larvae during the simulated heatwave and post-heatwave event. Sediment trap data indicated elevated mortality, particularly dominated by polychaete larvae during the post-heatwave period. Oxidative stress biomarker (lipid peroxidation LPX) levels in the plankton community correlated with temperature, signaling cellular damage during the heatwave. LPX increased and proteins decreased with increasing salinity during the experiment. Offspring production peaked during the post-heatwave phase. Notably, the calanoid copepod *Acartia clausi* exhibited a preference for ciliates as its primary prey, constituting 20% of the overall available prey. Our findings suggest a potential shift in coastal zooplankton communities during future marine heatwaves, transitioning from calanoid mesozooplankton dominance to a system featuring meroplankton and/or harpacticoid copepods. Although species preying on microzooplankton may gain advantages in such conditions, the study underscores the damaging impact of heatwaves on organismal lipids, with potential consequences for reproduction, growth, and survival within marine ecosystems.

funding from the European Union's Horizon 2020 Research and Innovation Program (H2020/2017– 2020) under grant agreement nr. 731065

**Competing interests:** The authors have declared that no competing interests exist.

## Introduction

Marine heatwaves are predicted to increase in frequency, intensity and duration in the upcoming future [1], posing a serious threat for the marine [1, 2]. Historically, especially Australian waters, the eastern Pacific Ocean and the Mediterranean Sea, and to some extent also the Atlantic Ocean and the Indian Ocean are areas with high vulnerability to marine heatwaves [3–7]. Ecosystem effects can persist for years after a prolonged oceanic heatwave [2, 8], with effects reaching from microbes to marine mammals, jeopardizing ecosystem sustainability. Vast literature shows that heatwave phenomena affect multiple marine taxa, such as polychaetes, kelps, pearl oysters, pteropods, gastropods, corals, fish, squid and crustaceans [9–15].

Elevated temperature can change the community structures to an increased abundance of species of smaller size, young age classes and a reduction in size-at-age [16]. These findings are verified by multiple studies; in general, large copepods decrease and small copepods increase in abundance after marine heatwaves [17]. Microzooplankton show a strong positive response to warming [18, 19], but elevated temperature increases the energy demand, and can increase foraging by mesozooplankton upon microzooplankton, reducing their biomass [20]. A large variability among results can also be observed; for example [21] measured decreased survival of the copepod *Pseudodiaptomus incisus* during heatwave exposure. On the other hand, transgenerational exposure to marine heatwaves fully mitigated its lethal effect. Increased survival came with a cost of reproduction, constrained by decreased grazing. Evans et al. [6] report that copepods, cladocerans and appendicularians suffered from marine heatwaves, whereas jellyfish, echinoderm larvae and bryozoans benefitted from heatwaves.

The Mediterranean Sea is an example of a semi-enclosed sea experiencing rapid warming, which is considered an amplified precursor of the changes to expect in the greater oceans. Studying the relationship between plankton and marine heatwaves in the Mediterranean Sea is critical due to the foundational role plankton play in marine ecosystems and their sensitivity to temperature changes. Changes in plankton dynamics can affect biogeochemical cycles, including carbon sequestration, thus influencing broader climate regulation processes [22]. The Mediterranean Sea's semi-enclosed nature and high biodiversity make it a crucial region for understanding these impacts. Researching plankton responses to marine heatwaves in this area can provide insights into the resilience and adaptability of marine ecosystems under climate change pressures and inform about conservation and management strategies [23].

The Thau lagoon has been in focus of extensive research due to its central role as a provider of important ecosystem services [24], and sensitivity to eutrophication, following limited water exchange. During recent years the lagoon has suffered heavily from heatwaves, which are getting more frequent in the area [25]. Mesocosm campaigns studying heatwave impact, and biogeochemistry in plankton communities have been executed in the Thau lagoon since 2006 [26–32]. Courboulès et al. [29] demonstrated that warming caused a rise in nanophytoplankton abundance, as well as in bacteria, cyanobacteria and viruses. Functional types/traits of phytoplankton are affected by heatwave, as diatoms, cyanobacteria, chlorophytes and prymnesiophytes were benefitted, but dinoflagellates can decrease [31].

Heatwaves are stressful for organisms and can cause physiological changes in cells. This is because the metabolism of organisms rises with increasing temperature which, on the other hand, produces more reactive oxygen species (ROS) that may lead to an imbalance between oxidants and anti-oxidants [33]. If cellular defense against ROS is not effective enough, longterm stress can have consequences for animal growth, survival, and reproduction [34]. This is because ROS can lead to cellular damage, such as DNA damage and lipid peroxidation LPX. The ROS can, however, be converted to less harmful molecules by different enzymes; for

example catalase CAT acts against ROS, and glutathione-s-transferase GST deactivates secondary metabolites [35 and references therein].

Heatwaves can cause significant stress on zooplankton organisms and can substantially shorten the lifespan of calanoid copepods by up to 50% [36]. This might be due to increased metabolic demand as higher temperatures accelerate metabolic processes, leading to faster energy depletion and reduced lifespan [37]. Also, elevated temperatures can impair reproductive processes, reducing the number of viable offspring and overall population sustainability [38] and prolonged exposure to heat stress can cause cumulative physiological damage, leading to premature death [39]. This can lead to shifts in community structure and thus changes in carbon sequestration and disruption of the food webs. In this study we investigated the zooplankton community composition, vital rates (including grazing, production, survival, and oxidative stress), and the quantity and quality of zooplankton prey, assessing whether these factors were impacted by the heatwave. The data were collected during a 20-day *in situ* mesocosm field experiment at a coastal infrastructure set up in Thau lagoon, southern France during an AQUACOSM-plus Transnational Access campaign (www.aquacosm.eu), studying plankton ecology and physiology using realistic temperature increase (heatwave and post-heatwave periods).

## Material and methods

### Study site

The mesocosm field experiment studying heatwave effects on the plankton community was conducted using the MEDIMEER platform (Mediterranean platform for Marine Ecosystems Experimental Research) based at the marine station of Sète (SMEL, University of Montpellier 2, 43˚ 24' 49" N, 3˚ 41' 19" E) in summer 2019. The experiment was conducted in the Thau lagoon in the western Mediterranean Sea which is oligotrophic [25] and is connected to the sea via two permanent inlets. Thau lagoon has extensive shellfish farming (provides 10% of the oysters in France; [24]) and the lagoon is also in frequent recreational use. The ecosystem has been extensively studied, both in the field, using experimental set-ups and by analysing long-term monitoring data (e.g., [25, 31]). The depth varies on average between 3 and 5 meters [40] and salinity has varied over the years between $\sim$ 30 and 40 [25].

### Mesocosm set-up

Technical details of the platform and information on data monitoring of infrastructure are reported by Nouguier et al. [41], Vidussi et al. [26], and Mostajir et al. [27]. The heatwave experiment was conducted between 24 May and 12 June 2019, during a total of 20 days. The size of the transparent mesocosm bags was 280 cm (length) and 120 cm (width), made of vinyl acetate polyethylene film (thickness: 200 μm), reinforced with nylon (Engineering Agency Haikonen Ky), and equipped with a 50 cm long sediment trap. Mesocosms were covered with a polyvinyl-chloride dome, which transmitted 73% of the photosynthetically active radiation PAR, to avoid external inputs and precipitation. A pump (Rule, Model 360) was immersed at 1 m depth to gently mix the mesocosm water.

Two treatments, each in triplicate, were applied to six mesocosms. During the first 10 d of the experiment (D1–D10), the water temperature was raised +3˚C in three mesocosms (heatwave H), while the three remaining mesocosms had natural lagoon temperature (control C). After this 10 days period (d11–d20), the temperature was returned to ambient water temperature (post-heatwave PH). During this period, the "control mesocosms" experienced the *in situ* lagoon temperature. In addition, two incubation mesocosms with the same water temperature

as Control and Heatwave experimental mesocosms were established to incubate several samples, thereby avoiding contamination of the experimental mesocosms.

Detailed information on the installed high-frequency sensors for measuring temperature, chlorophyll *a*, conductivity, and light, as well as results on these data and nutrients are found in Soulié et al. [31]. In brief, in each experimental mesocosm, a set of automated high-frequency sensors was immersed at a depth of 1 m. Each set consisted of an oxygen optode (Aanderaa 3835) for measuring dissolved oxygen concentration and saturation, a chlorophyll fluorometer (WetLabs ECO-FLNTU) for chlorophyll *a* fluorescence, an electromagnetic induction conductivity sensor (Aanderaa 4319) for salinity, and a spherical underwater quantum sensor (Li-Cor Li-193) for incident photosynthetically active radiation. Moreover, three water temperature probes (Campbell Scientific Thermistore Probe 107) were installed at three depths (0.5, 1 and 1.5 m) of each mesocosm. Measurements were taken at every minute during the entire experiment.

## Zooplankton community composition, copepod feeding and production

Zooplankton abundance and community composition were monitored at the time of the filling (day 0, hereafter D0) and at the end by emptying mesocosms (D20). During the filling of the enclosures, three different volumes of seawater ($2 \times 50$ l, $1 \times 122$ l) from ambient lagoon water were filtered through a 200 μm WP2 net using a submersible pump. During the emptying of each mesocosm at the end of the experiment, a volume of approximately 300 l from both control and heatwave mesocosms was filtered through the same zooplankton net, as mentioned above. In addition, 15–20 l of water was collected during D6, D12 and D18 and filtered onto a 150 μm mesh net and samples were preserved in buffered-formaldehyde (4% final concentration) for determination of mesozooplankton community composition.

The sediment traps were sampled every four days (D3, D7, D11, D15 and D19) by closing the trap connection to the mesocosm, removing it and replacing the trap with a new one. Then 10–20 mL subsamples from the traps (in total 30 samples) were preserved and then analysed for the determination of mesozooplankton abundance and community composition.

All samples were preserved in 4% buffered-formaldehyde seawater solution until taxonomic analysis was carried out. Taxonomic identification and counts of zooplankton were performed in the laboratory using an Olympus SZX12 dissecting microscope. Abundance was expressed as the number of individuals per cubic meter (ind. $m^{-3}$), whereas relative abundance (%) refers to the evenness of distribution of individuals among species in a community.

Copepod feeding experiments were performed three times in total: at the beginning prior to the increasing temperature (D1), at the end of the heatwave event (D9) and at the end of the post-heatwave event (D19) for both Control and Heatwave mesocosms. Copepods were collected from the surrounding area by oblique tows, using a boat, and a 200 μm WP-2 net equipped with a non-filtering cod-end. Healthy females of dominant copepod species *Acartia clausi* were sorted under a stereomicroscope (Olympus SZX10) and acclimated in 2 l polycarbonate bottles added with pre-filtered mixed subsurface seawater from one mesocosm of each treatment (C1 and T1). After the acclimation (18–24 h), the copepods were transferred to new seawater from the same mesocosms and re-incubated for 24h in the Incubation Mesocosms. In the beginning and at the end of the incubation, the water was sub-sampled for assessment of pigments using High-performance liquid chromatography (HPLC) (1 l) and microplankton communities (200 ml) using reverse filtration through a 150 μm mesh. The incubated animals were collected, checked for mortality and their prosome length was measured. The samples for microplankton enumeration (ciliates and dinoflagellates) were preserved in 2% acid Lugol's solution, settled in sedimentation chambers (50 ml aliquots) and counted under an inverted

microscope (Olympus CK2). Cell volumes were estimated assuming simple geometric shapes and converted to biomass using the biovolume-carbon conversion for ciliates of 0.19 pg C $mm^3$ [42] and the carbon to volume relationships for dinoflagellates [43]. The carbon content of *Acartia clausi* was estimated from length measurements applying the length-body carbon regression equations by Christou and Verriopoulos [44]. Copepod clearance rates on phytoplankton pigments, ciliates, and dinoflagellates were calculated according to Frost [45]. Ingestion rates were calculated by multiplying clearance rates by the initial standing stocks [46]. The % of daily body carbon ingested was calculated using the mean female carbon weight of the respective copepod species. Subsamples for pigment analysis (1 l) were filtered on GF/F glass fibre filters using a low-vacuum pump then frozen in liquid nitrogen and stored at -80˚C until analysis. Pigments (chlorophylls and carotenoids) were extracted using 2 ml of methanol 95% as described in Vidussi et al. [26] and analyzed using the method of Zapata et al. [47], following the protocol and the HPLC system (Waters) detailed in Vidussi et al. [26]. Chemotaxonomic assignment of pigments followed that used in Soulié et al. [31], notably fucoxanthin was mainly assigned to diatoms while chlorophyll *a* was used as a proxy of total phytoplankton biomass.

Samples for copepod egg and naupliar standing stocks in the mesocosms were taken every second day (D1, D3, D5, D7, D9, D11, D13, D15, D17 and D19) according to the methodology suggested by Pitta et al. [48]. Approximately 5 l of water from the mesocosms was screened onto a 53 μm mesh net and samples were preserved in formalin (4% final concentration). Eggs and nauplii were enumerated under an Olympus SZX12 dissecting microscope. In the same samples, zooplankton community composition was identified and counted.

## Zooplankton biomarkers

To see how different copepod communities are affected by and recover from heatwave, the level of antioxidant defence (glutathione s-transferase GST and catalase CAT activities) and oxidative stress marker (lipid peroxidation LPX) were measured. The community samples were snap-frozen in liquid nitrogen, and stored in -80˚C, and shipped on dry-ice to Finland. For assays, the community samples were homogenized in 75 μl of 0.1 M $K_2HPO_4$ + 0.15 M KCl buffer (pH 7.4) using a Tissue Lyser II bead mill (Qiagen, Hilden, Germany), 2 × 2 min. at 30 shakes s$^{-1}$ after adding two stainless steel beads (∅ 1 mm) to each community vial. Subsequently, 25 and 5 μl of raw homogenate were aliquoted in 1.5 ml Eppendorf microtubes, snap-frozen in liquid nitrogen, and successively stored at -80˚C for LPX determination and bicinchoninic acid BCA protein assay, respectively. The remaining homogenate sample was centrifuged at 10,000g at +4˚C for 15 min. and supernatant was collected and aliquoted in different tubes for other assays. The leftover supernatant was stored at -80˚C for back up experiments in case of problems with original samples. The total protein concentrations were determined with a BCA Protein Assay kit (Thermo Scientific, Rockford, IL, USA) using serial dilution of bovine serum albumin (1–6 mg ml$^{-1}$) as a standard (Thermo Scientific, Rockford, IL, USA). Spectrophotometric measurements of the protein levels were performed at 570 nm using a Wallac EnVision 2103 Multilabel Reader (Perkin Elmer, Turku, Finland). LPX was measured using the ferrous oxidation-xylenol orange method (FOXII), following the protocol described in Vuori and Kanerva [49]. For this, 25 μl of the homogenate was diluted in 90 μl of methanol, the supernatant was collected and diluted again in methanol to reach a final volume of 100 μl. The samples were incubated in the dark for 2h at room temperature with 900 μl of reaction mix (one part of 2.5 mM ammonium iron (II) sulfate in 0.25 M $H_2SO_4$ + 9 parts of 0.111 mM xylenol orange in methanol). The absorbance was measured at a wavelength of 590 nm with a Wallac EnVision 2103 Multilabel Reader (Perkin Elmer, Turku, Finland). The LPX was

calculated per mg protein in the community sample. The GST was determined based on the conjugation of reduced glutathione GSH to 1-chloro-2,4-dinitrobenzene CDNB ([50], modified according to Vuori et al. [51]). Briefly, 3 μl of the sample was mixed with 47 μl of reaction mix (98% 0.1M phosphate-buffered saline PBS and 1% 200 mM GSH + 1% 100 mM CDNB) in a 384- well plate and absorbance was read at 340 nm for 6–13 min. CAT dissociates hydrogen peroxide $H_2O_2$ to water and dissolved oxygen $O_2$, and its activity was measured using an UV spectrophotometric method [52], monitoring the change of 240 nm absorbance after homogenate was mixed with $H_2O_2$ solution. For the assays, 2 μl of community sample (diluted at 0.3 mg ml$^{-1}$) was mixed with 13 μl of assay buffer (500 mM potassium phosphate buffer, pH 7.0). The assays were performed according to the instructions of the Catalase Assay Kit (Sigma-Aldrich, Merck KGaA, Darmstadt, Germany). Both the GST and CAT activities were calculated per mg of protein in the community sample. All samples, standards, and blanks were analyzed in triplicate. A mean coefficient variation percentage (CV%) was applied to each triplicate and those >10 were discarded from the analyses.

## Statistical analyses

General Linear Models (GLM) in SPSS version 28.0 were used to analyze differences between Treatment (heatwave + post-heatwave) and Control for mesocosm data of offspring production and grazing, and community composition in water and sediment traps.

We used linear mixed models (LMMs), based on restricted maximum likelihood RML estimation, for comparing biomarker levels of plankton community and for analysing effects of heatwave and salinity. Oxidative stress and anti-oxidant enzymes were treated as response variables, environmental variables (heatwave and salinity) as fixed effects, and the mesocosms as random effect. All response variables were tested for normality and homoscedasticity. For LMMs, the package *lmerTest* was used in the free software R, version 3.4.3 (www.R-project.org).

One-way ANOVA was performed with STATGRAPHIC software, to test differences in different copepod potential prey during the grazing experiments among treatments (controlled and heated), followed by a post-hoc Tukey's test (α = 0.05).

Zooplankton abundance was correlated against time using Pearson's correlation analysis. Principal Component Analysis (PCA) was used to evaluate potential relationships between the responses to the heatwave of different zooplankton taxa abundance (Cyclopoida, Harpacticoida, Calanoida, Gastropoda, Polychaeta and Lamellibranchia), offspring production (OP) and sediment traps (ST), with environmental (temperature, salinity) and biological (chlorophyll *a*) variables. The responses to heatwave of zooplankton biomarkers (LPX, GST, CAT, protein) were analysed using the same variables as mentioned above, with PCA. All data were log transformed. Data analyses were performed using R (version 4.3.1) software.

## Results

### Experimental conditions

On the first day the water temperature was almost similar in both treatments (Control C: 18.57°C, Heatwave HW: 18.90°C), after which it was artificially elevated ∼3°C in the heated enclosures, and was kept 3°C above the control temperature for 10 days (= Heatwave; 21.3 ±0.4°C), after which the heating was ceased and the temperature decreased to ambient (= post-heatwave; 19.7±0.3°C), as in the control (19.8±0.3°C; Fig 1). Salinity showed some variation between the mesocosms and increased with time in all mesocosms (38.5±0.1), due to evaporation. Chlorophyll *a* (Chl *a*) from fluorescence measurements showed also some variability among the mesocosms (C: 1.28±0.50 μg l$^{-1}$, T: 1.27±0.51 μg l$^{-1}$). However, it decreased in the enclosures with time (until D13), and after D15 Chl *a* started to increase again (Fig 1).

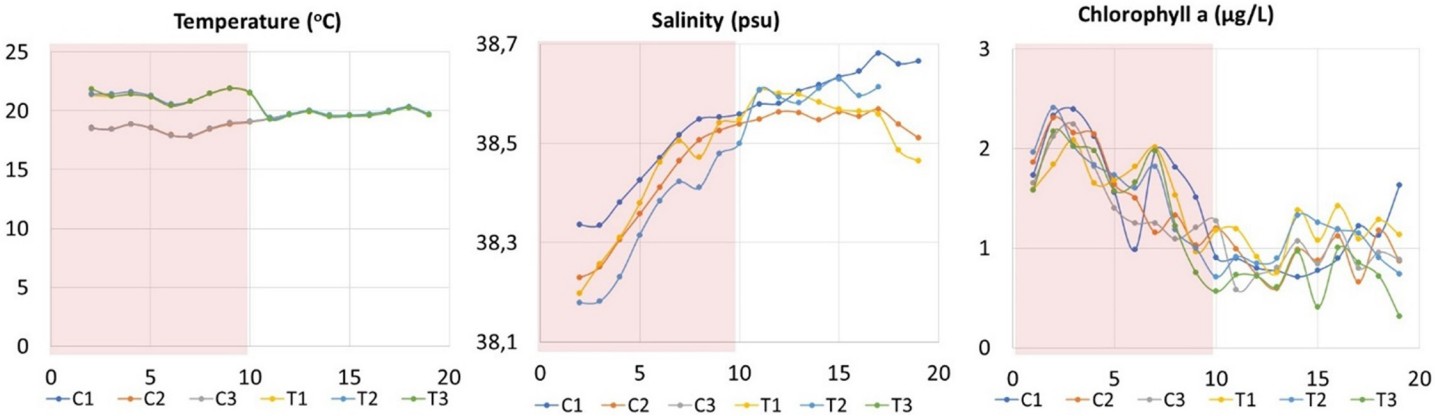

**Fig 1. Distribution of environmental conditions (Temperature, salinity and chlorophyll _a_ concentration) in the control (C1, C2, C3) and heated (T1, T2, T3) treatments.** The red shaded area represents the HW period during which the +3°C was applied in the HW mesocosms. Note that the scales of the y-axes are different.

## Zooplankton community composition

At the start of the experiment (D0), total zooplankton abundance in the Thau lagoon was 1227 ± 358 ind. m⁻³ and the community was dominated by Copepoda, Cladocera, Polychaeta larvae, Gastropoda larvae and Bivalvia larvae. During the experiment (D1-D19), in total, four groups dominated the communities in the mesocosms, i.e. copepods, larvae of bivalves, polychaetes and gastropods followed by other groups such as appendicularians, larvae of echinoderms, medusae and cladocerans that contributed with low numbers to the total zooplankton community (Fig 2). The copepod community in the mesocosms (Fig 3) consisted mainly of Harpacticoida (up to 80%), _Acartia clausi_ (up to 43%), _Oithona_ spp. (up to 29%). Other species like _Oncaea_ spp, _Isias clavipes_, _Centropages_ spp. and other unidentified juvenile Calanoida and Cyclopoida were also present in the mesocosms but their abundances were low (<10%). At the end of the experiment (D20), the zooplankton abundance in both control and treatment mesocosms had a mean of 4471±1198 and 4481±903 ind. m⁻³, respectively, and the zooplankton community was dominated mainly by Copepoda and the meroplankton larvae of Polychaeta, Cirripedia and Gastropoda (Fig 2).

For zooplankton community composition in the mesocosm water, we found a significant positive effect of the treatment i.e., heatwave and post-heatwave vs. control, on total copepod abundances in the mesocosms (p = 0.026; Table 1), due to increasing harpacticoid copepods during post-heatwave (p = 0.019). The other copepod taxa, cyclopoid and calanoid copepods did not vary in abundance between control and treatment (p > 0.05). The differences in copepod abundances were mostly due to harpacticoid copepods increasing during post-heatwave. For gastropod larval abundances, there were no significant changes over time in the mesocosms. For polychaete larval abundances, we detected a trend (p = 0.088) between the control and treatment, i.e., heatwave/post-heatwave (Table 1). The results can be explained by high numbers of polychaete larvae during post-heatwave.

Zooplankton abundance found in the sediment traps changed with time in all mesocosms (Fig 4). Total zooplankton abundance increased in the sediment traps in the course of the experiment, both in the control and the heatwave treatment, showing that the zooplankton mortality was higher in the heatwave treatment (Pearson's correlation, R = 0.575, N = 30, p < 0.001). Moreover, variations of different taxa abundances in the sediment trap samples were studied and analyzed in the control and heatwave treatment. Results showed that Polychaeta larvae in the sediment traps did not vary significantly between treatments (Table 1), and neither did the

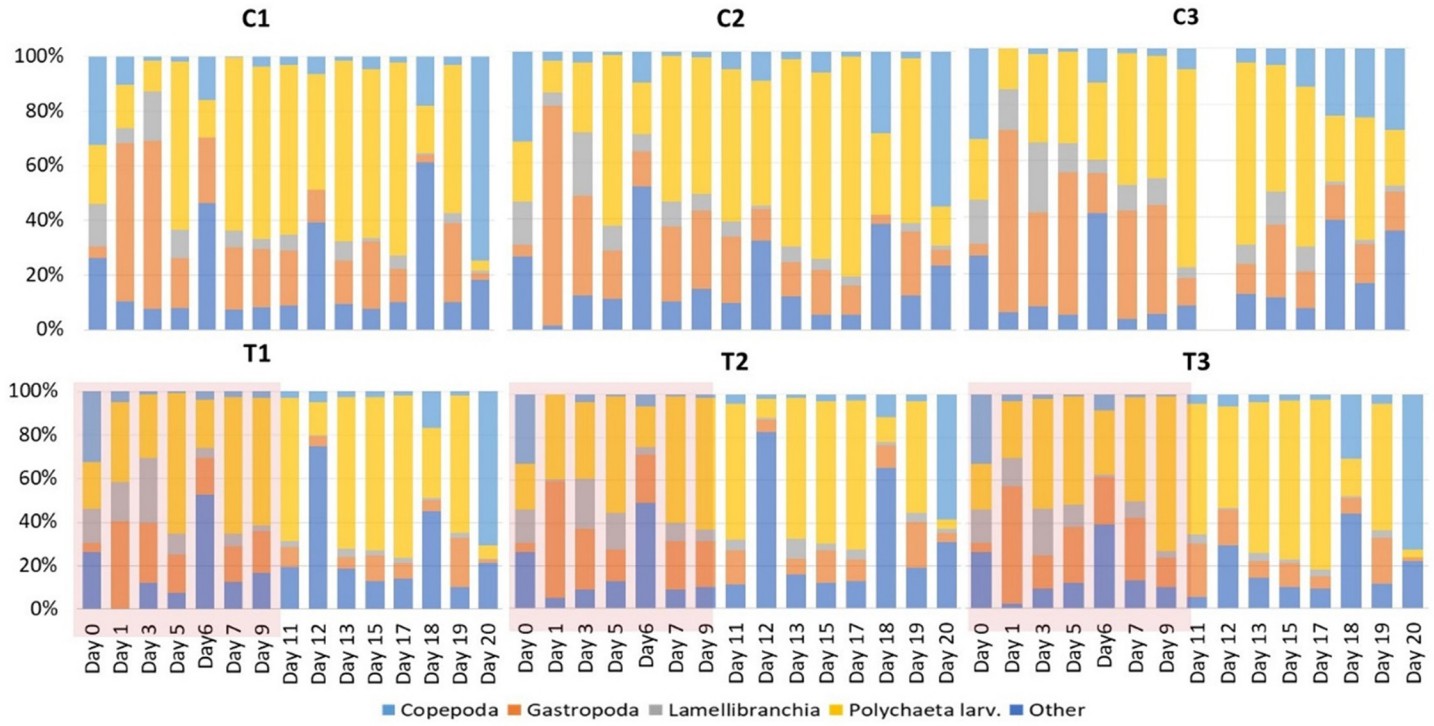

**Fig 2. Distribution of relative abundance (%) of zooplankton groups sampled from the water column of control (C1, C2, C3) and heated (T1, T2, T3) mesocosms.** The red shaded area represents the HW period during which the +3°C was applied in the HW mesocosms.

other taxa. The copepod community showed some differences as calanoid copepods in sediment traps were significantly higher in heatwave than in the control (Table 1). Other taxa abundances in sediment traps (copepodites, Harpacticoida and Cyclopoida) were not significantly different between control, treatment heatwave and post-heatwave in sediment traps.

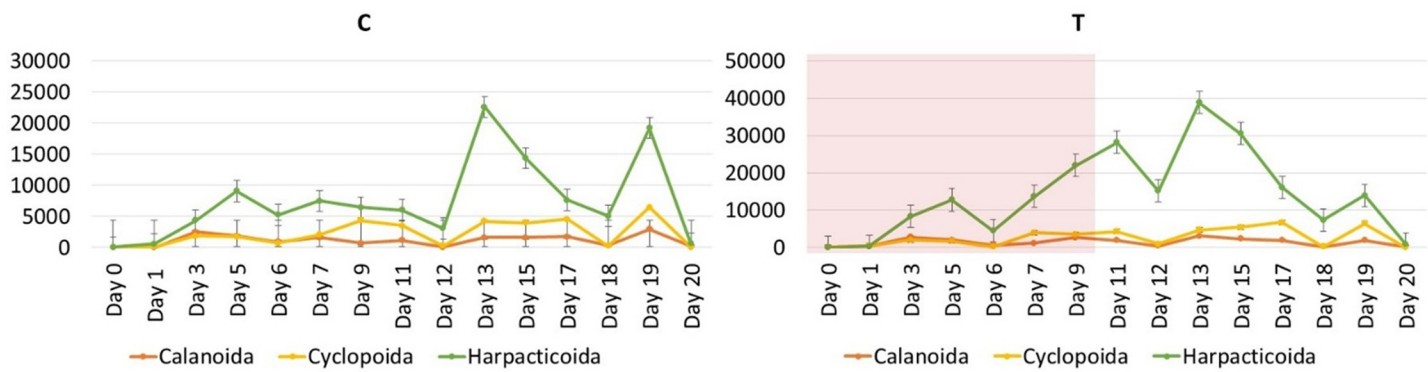

**Fig 3. Distribution of mean abundance (ind. m$^{-3}$ with SD) values of three main copepod families (Calanoida, Cyclopoida, Harpacticoida) sampled from the water column of control (C) and heated (T) mesocosms.** The red shaded area represents the HW period during which the +3°C was applied in the HW mesocosms.

**Table 1. General linear model results for zooplankton grazing, production and community composition in water and sediment traps, and linear mixed model for biomarkers.** Time period is heatwave, and post-heatwave; the control was not treated with increased temperature. NS = not significant, CR = clearance rates, IR = ingestion rates, D = day. As random factor (RF) in LMM was used 1 | mesocosm enclosures.

|  | Dependent variables | RF variance | factors | Value ± S.E. | F or t | df | p |
|---|---|---|---|---|---|---|---|
| Community<br>• water | copepods | | treatment | | 5.2 | 1,59 | = .026 |
| | gastropods | | | | 1.7 | 1,59 | NS |
| | polychaetes | | | | 3.0 | 1,59 | = .088 |
| • sediment traps | *calanoid copepods | | treatment | | 7.6 | 1,29 | < .01 |
| | polychaetes | | | | .03 | 1,29 | NS |
| | gastropods | | | | .23 | 1,29 | NS |
| | lamellibranchia | | | | .16 | 1,29 | NS |
| Production | offspring | | | | 4.7 | 1,59 | = .034 |
| Biomarkers | LPX | 0.03 | salinity | 15.8±7.8 | 2.014 | 73.9 | .0477 |
| | CAT | 5741 | time period | 4.3±2.2 | 1.925 | 72.4 | .0581 |
| | GST | $2.6 \times 10^{-16}$ | salinity | 314.9±579.4 | .29 | 46.2 | NS |
| | protein | .0 | time period | 596.4±172.5 | 11.9 | 46.7 | .00117 |
| | | | salinity | 0.4±0.3 | 68 | 1.4 | NS |
| | | | time period | -.09±.08 | 68 | -1.2 | NS |
| | | | salinity | -8.8±3.5 | 75 | -2.5 | .0133 |
| | | | time period | 1.5±0.97 | 75 | 1.6 | NS |
| Grazing | Dinoflag (CR) (IR) | | D19 | | 14.5 | 1,6 | = .019 |
| | chl *a* | | D19 | | 28.9 | 1,6 | = .006 |
| | ciliates (CR) | | D19 | | 4.3 | 1,12 | NS |
| | (IR) | | D9, D19 | | 10.9 | 1,6 | = .06 |
| | | | D9, D19 | | | | = .09 |

*other analysed copepod taxa were not significant (Cyclopoida, Harpacticoida, Copepodites)

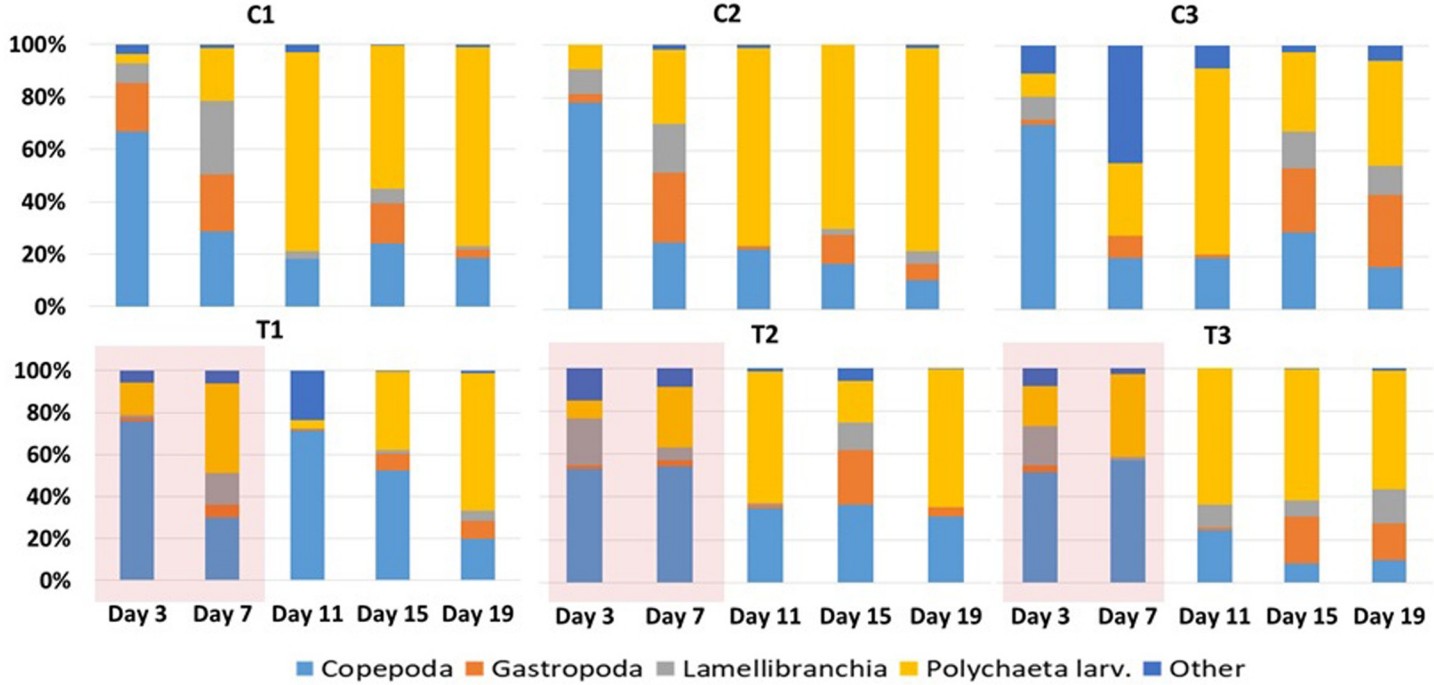

**Fig 4. Distribution of relative abundance (%) of zooplankton groups sampled from sediment traps of control (C1, C2, C3) and heated (T1, T2, T3) mesocosms.** The red shaded area represents the HW period during which the +3˚C was applied in the HW mesocosms.

## Offspring production

Offspring (eggs and nauplii) production of copepods varied between treatments (p = 0.034; Fig 5), showing the number of offspring was higher in the treatment (post-heatwave) (Table 1). Offspring number was quite similar during the heatwave and in the control between D1 and D7 (C: 10 ± 5; T: 10 ± 6 offspring l$^{-1}$), whereas the higher numbers of offspring were found during D9, D11 and D13 (57 ± 47 offspring l$^{-1}$) in the heated mesocosms. In the control mesocosms, offspring production (34 ± 9 offspring l$^{-1}$) were found higher at the end of the experiment (D19).

## Copepod grazing

Initial chlorophyll $a$ concentrations were low during the individual grazing experiments and varied between 1.55 ± 0.14 µg l$^{-1}$ (T, D19) and 2.62 ± 0.29 µg l$^{-1}$ (C, D9) (Table 2). Diatom pigment concentrations (fucoxanthin) were similar during the experiments in D1 and D9 both in control and heat treatments, and varied from 1.18± 0.03 µg l$^{-1}$ (C, D 9) to 1.46 ± 0.35 µg l$^{-1}$ (T, D9), whereas at the end of the experiment (D19) was almost 1.5 to 1.8 times higher in the heat treatment (2.19±0.28 µg l$^{-1}$) (Table 2). Dinoflagellates contributed significantly (83%) to the total heterotrophic biomass grazed by copepods, mainly during the first days of the experiment and varied from 1.33 ± 0.74 (T, D19) to 36.92 ± 2.26 mg C l$^{-1}$ (C, D1). Ciliates (17%) comprised a minor fraction of the total heterotrophic grazed biomass and varied between 0.42±0.16 (C, D9) and 7.99±2.15 mg C l$^{-1}$ (C, D1). The biomass of the above prey was significantly different among the different treatments and days (ANOVA, p < 0.0001). Clearance rates by *Acartia clausi* were the highest on ciliates (max. 135.7 ± 58.7 ml ind$^{-1}$ d$^{-1}$, C-D9) and on dinoflagellates (max. 41.85 ± 9.29 ml ind$^{-1}$ d$^{-1}$, C-D19), whereas the lowest clearance rate was found on

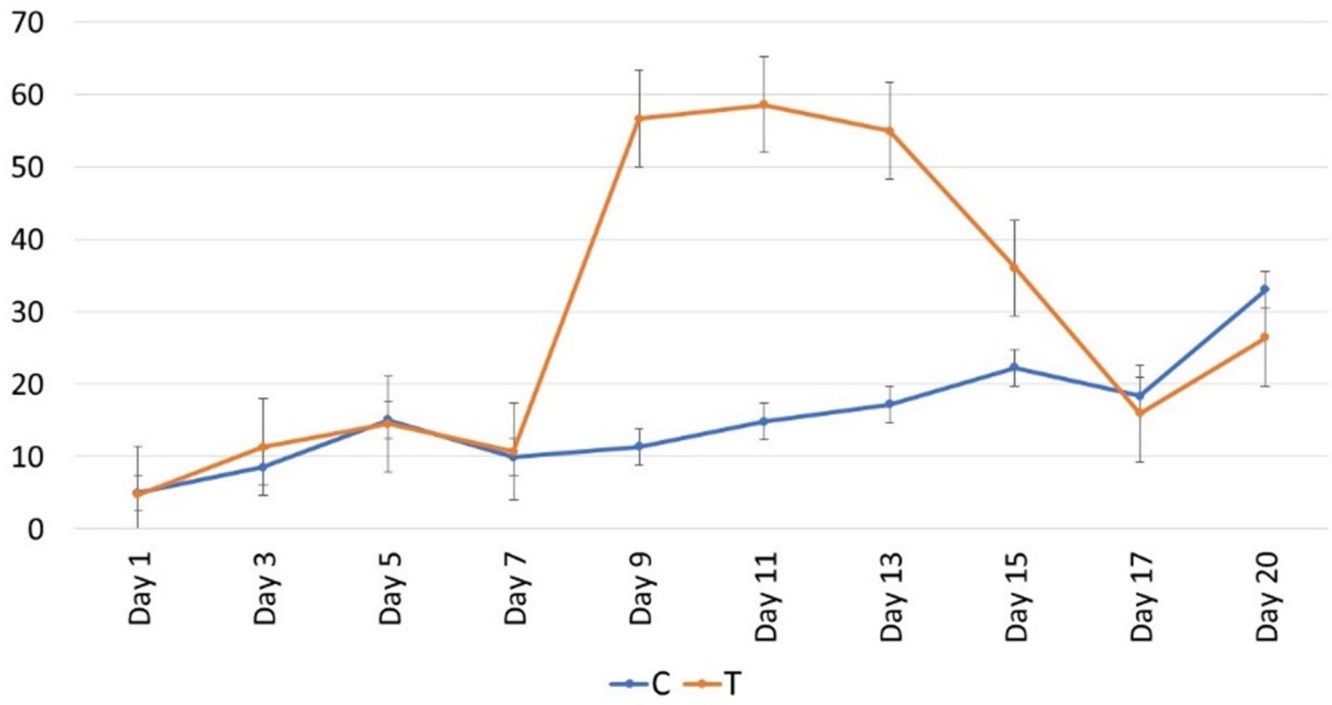

**Fig 5. Temporal variation of mean offspring production values (eggs+nauplii per liter with SD) of control (C) and heated (T) mesocosms.** The red shaded area represents the HW period during which the +3°C was applied in the HW mesocosms.

**Table 2. Initial concentrations of total chlorophyll *a*, diatoms and biomass of heterotrophic prey (dinoflagellates and ciliates) converted to carbon biomass (mgC l$^{-1}$), clearance rates (ml cop$^{-1}$ day$^{-1}$) and specific ingestion (% body C ingested day$^{-1}$) of *Acartia clausi* on chlorophyll *a*, diatoms, dinoflagellates and ciliates during the incubation experiments in control C and heat T treatments on Days 1, 9 and 19.** Error bars indicate± STDV of the mean.

| | Treatment | Day | Chlorophylla (mg l$^{-1}$) | Diatoms (mg l$^{-1}$) | Dinoflagellates (mgC l$^{-1}$) | Ciliates (mgC l$^{-1}$) |
|---|---|---|---|---|---|---|
| *Initial prey* | | | | | | |
| | C | D1 | 2.52 ± 0.08 | 1.70 ± 0.07 | 36.92 ± 2.26 | 7.99 ± 2.15 |
| | | D9 | 2.63 ± 0.30 | 1.19 ± 0.03 | 17.43 ± 2.03 | 0.42 ± 0.16 |
| | | D19 | 1.55 ± 0.14 | 1.38 ± 0.12 | 13.77 ± 3.7 | 3.74 ± 1.47 |
| | T | D1 | 2.41 ± 0.08 | 1.46 ± 0.27 | 29.67 ± 4.05 | 6.16 ± 1.59 |
| | | D9 | 1.91 ± 0.16 | 1.47 ± 0.35 | 21.25 ± 1.33 | 0.67 ± 0.06 |
| | | D19 | 1.59 ± 0.31 | 2.20 ± 0.29 | 8.00 ± 0.74 | 3.11 ± 0.78 |
| **Clearance rates (ml ind$^{-1}$ day$^{-1}$)** | | | | | | |
| | C | D1 | 7.26 ± 2.76 | 3.70 ± 0.19 | 29.40 ± 5.76 | 28.11 ± 10.55 |
| | | D9 | 26.37 ± 10.28 | 32.17 ± 12.75 | 29.67 ± 10.98 | 99.26 ± 32.36 |
| | | D19 | 11.7 ± 2.27 | 6.29 ± 2.40 | 41.85 ± 9.29 | 83.79 ± 28.07 |
| | T | D1 | 3.29 ± 0.33 | 3.49 ± 1.47 | 13.38 ± 2.37 | 29.52 ± 15.83 |
| | | D9 | 0.15 ± 0.01 | | 13.56 ± 7.03 | 135.75 ± 58.70 |
| | | D19 | 18.30 ± 10.13 | 3.73 ± 1.00 | 12.22 ± 5.75 | 40.14 ± 28.38 |
| **% body carbon ingested day$^{-1}$** | | | | | | |
| | C | D1 | 62.73 ± 38.61 | 33.49 ± 18.51 | 60.74 ± 6.25 | 7.82 ± 3.53 |
| | | D9 | 39.40 ± 13.91 | 90.90 ± 54.31 | 36.54 ± 15.84 | 3.90 ± 1.70 |
| | | D19 | 58.17 ± 17.00 | 34.20 ± 17.79 | 34.31 ± 6.23 | 7.05 ± 0.61 |
| | T | D1 | 27.40 ± 19.38 | 26.24 ± 18.56 | 25.66 ± 2.55 | 8.32 ± 3.48 |
| | | D9 | 0.75 ± 0.13 | | 16.45 ± 8.31 | 2.68 ± 1.27 |
| | | D19 | 84.92 ± 26.69 | 21.08 ± 11.14 | 7.82 ± 1.82 | 4.02 ± 1.43 |

chlorophyll *a* (3.29 ± 0.33 ml ind$^{-1}$d$^{-1}$ T-D1) (Fig 4). Daily rations varied depending on the prey type and the maximum ingestion was found for diatoms (91 ± 54%) (Table 2).

GLM analysis was performed on copepod clearance and ingestion rates separately for chlorophyll *a*, dinoflagellates, and ciliates. Diatoms were not included for the analysis due to lack of data. For dinoflagellates, both clearance rates and ingestion rates of post-heatwave D19 were significantly higher in the control than in the heat treatment. For chlorophyll *a*, there were no significant differences between control and post-heatwave D19 (p > 0.05). For ciliate prey, there was a significant trend in both copepod clearance rates and ingestion rates, analysed as between D9 and D19 (Table 1).

## Zooplankton biomarkers

The result for zooplankton community lipid peroxidation (LPX) showed that the levels were high during heatwave period, and started to increase during post-heatwave period, as the treatment showed a significant trend (Table 1, Fig 6). Salinity also affected LPX, i.e., cell damage, measured as lipid peroxidation, as LPX increased significantly with salinity levels (Table 1).

For catalase CAT, treatment was significant (Table 1), indicating CAT levels were higher towards the end of the experiment (= post-heatwave). Salinity did not have a significant effect on CAT enzyme levels.

Protein levels in the zooplankton community did not differ between treatments, whereas the salinity effects were significant (Table 1), as levels decreased with increasing salinity. Glutathione-s-transferase GST in the zooplankton community showed neither any significant

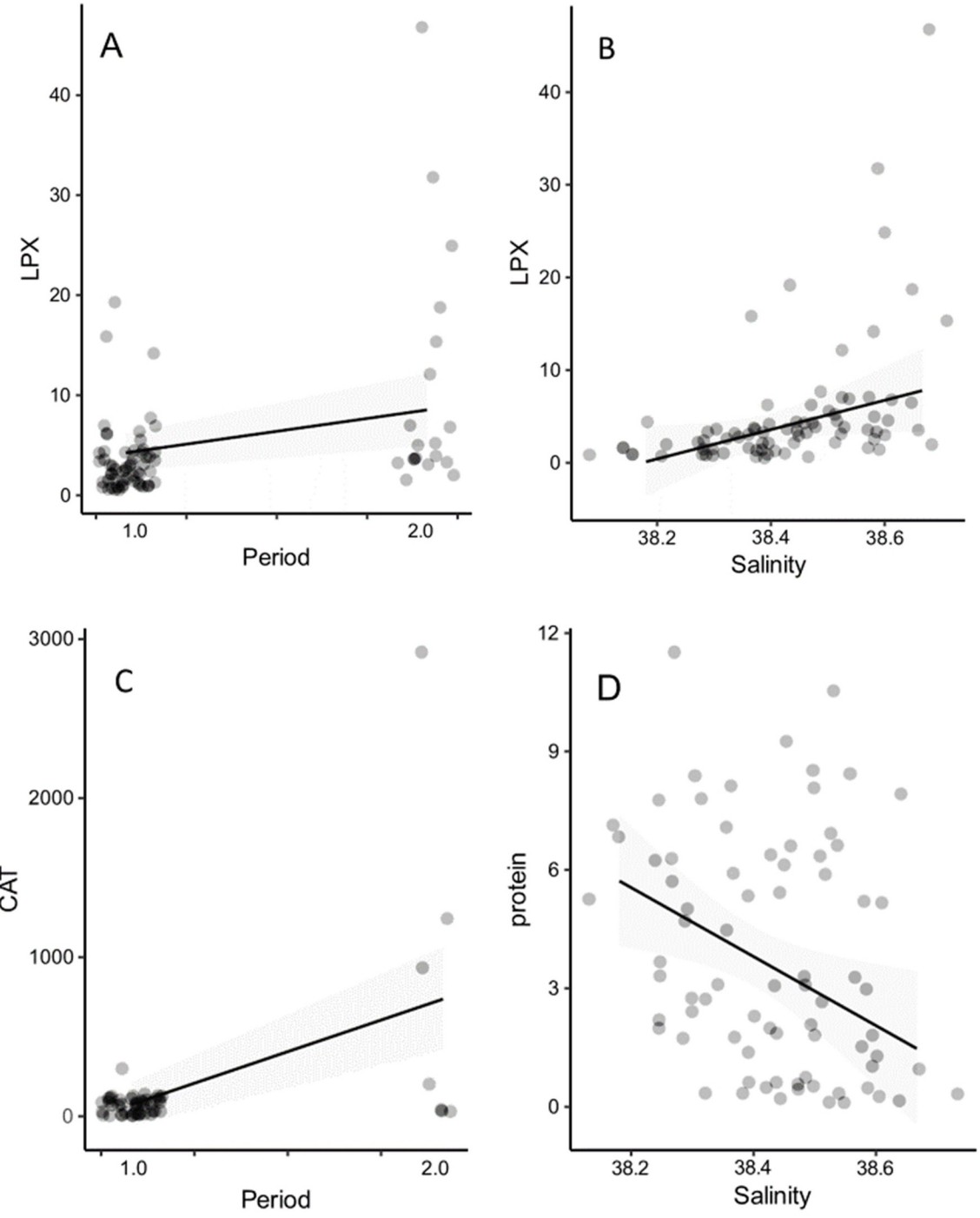

**Fig 6. Biomarker enzymes measured from >200 um plankton community during heatwave experiment in Control C and Treatment T, consisting of 10 days heatwave following 10 days of post-heatwave.** A) Lipid peroxidation (LPX), FOX mg protein$^{-1}$ during Period 1 (Heatwave) and Period 2 (Post-heatwave), and B) LPX as a relationship with salinity, C) catalase (CAT) µmol min$^{-1}$ mg$^{-1}$, as well as D) protein mg$^{-1}$ against increasing salinity. FOX = ferrous oxidation - xylenol orange. Note salinity scale in Fig B and D.

differences in treatment and salinity. We also analysed samples of *Acartia clausi* copepod, but no differences were found as a response to elevated temperature ($p > 0.05$).

## Relationships between zooplankton taxa, sediment traps, biomarkers and environmental variables

Two PCAs were performed to highlight the potential correlations, firstly between zooplankton data (abundance of zooplankton taxa (Cyclopoida: CYC, Harpacticoida:HAR, Calanoida:CAL, Gastropoda:GAS, Polychaeta:POL, Lamellibranchia:LAM), offspring production (OP), total zooplankton abundance in sediment traps (ST)) and the environmental-biological (temperature, salinity, chlorophyll *a*) variables, and secondly between zooplankton biomarkers and the environmental-biological variables, both during the heatwave and post-heatwave periods (Fig 7 and Table 3).

In the PCA performed using zooplankton data and environmental-biological parameters, during the heatwave period the first and second components explain the variance at 68.3% and 11.6% respectively, whereas during the post-heatwave period the first two axes represented 77% of the total variance (Fig 7 and Table 3). For both periods, CYC, HAR, CAL, GAS, POL, LAM, ST and OP were clustered near the second PCA axis (Fig 7 and Table 3). Temperature was part of this group during the heatwave period while salinity was closer to the group during the post-heatwave (Fig 7 and Table 3). During the heatwave period, chlorophyll *a* was opposed to salinity suggesting a negative relationship between them. Conversely, during the post-heatwave chlorophyll *a* concentration was on the first PCA axis suggesting no effect to the rest of the parameters. Moreover, temperature was opposed to the rest of the parameters suggesting a negative relationship between them.

Elevated temperature or warming was closely associated with oxidative stress response, both during heatwave and post-heatwave. Anti-oxidant enzymes responded also to the heat; GST and CAT were tightly connected to temperature during post-heatwave.

Using all biomarker data, and dividing them into heatwave and post-heatwave, variation in dimension 1 was explained by 36.1% and 46.9% for heatwave and post-heatwave, respectively (Fig 7, Table 3). Dimension 1 was not clearly explained by any of the environmental variables in post-heatwave, perhaps chlorophyll *a* during heatwave. Dimension 2 (temperature) was explained by 23.1% and 17.8% in heatwave and post-heatwave, respectively.

## Discussion

The aim of the present study was to assess the effects of simulated heatwave on the plankton community in a coastal Mediterranean lagoon during late spring/early summer. A major finding of our study was the significant change in zooplankton from a copepod-dominated to a meroplankton-dominated community. Copepod offspring production was high during the post heatwave and the grazing rates revealed different prey-dependent responses between groups. Biomarkers such as lipid peroxidation, measured in the plankton community indicated cell damage occurrence, and were strongly correlated with temperature.

### Zooplankton community composition in the water and sediment traps of the mesocosms

Increased temperature affects the zooplankton community in the mesocosms, and this is in line with temperature being a major determinant for metabolic rate, body size, hatching success, development rate and /or fecundity of copepods (e.g. [53–55]). We demonstrated that harpacticoid copepods and polychaete larvae managed better in the +3˚C simulated heatwave than other zooplankton, as well as in the post-heatwave (during which temperature was allowed to return to natural ambient temperature) during a field *in situ* mesocosm experiment in Thau lagoon, southern France. Mortality in the zooplankton community was high, especially during

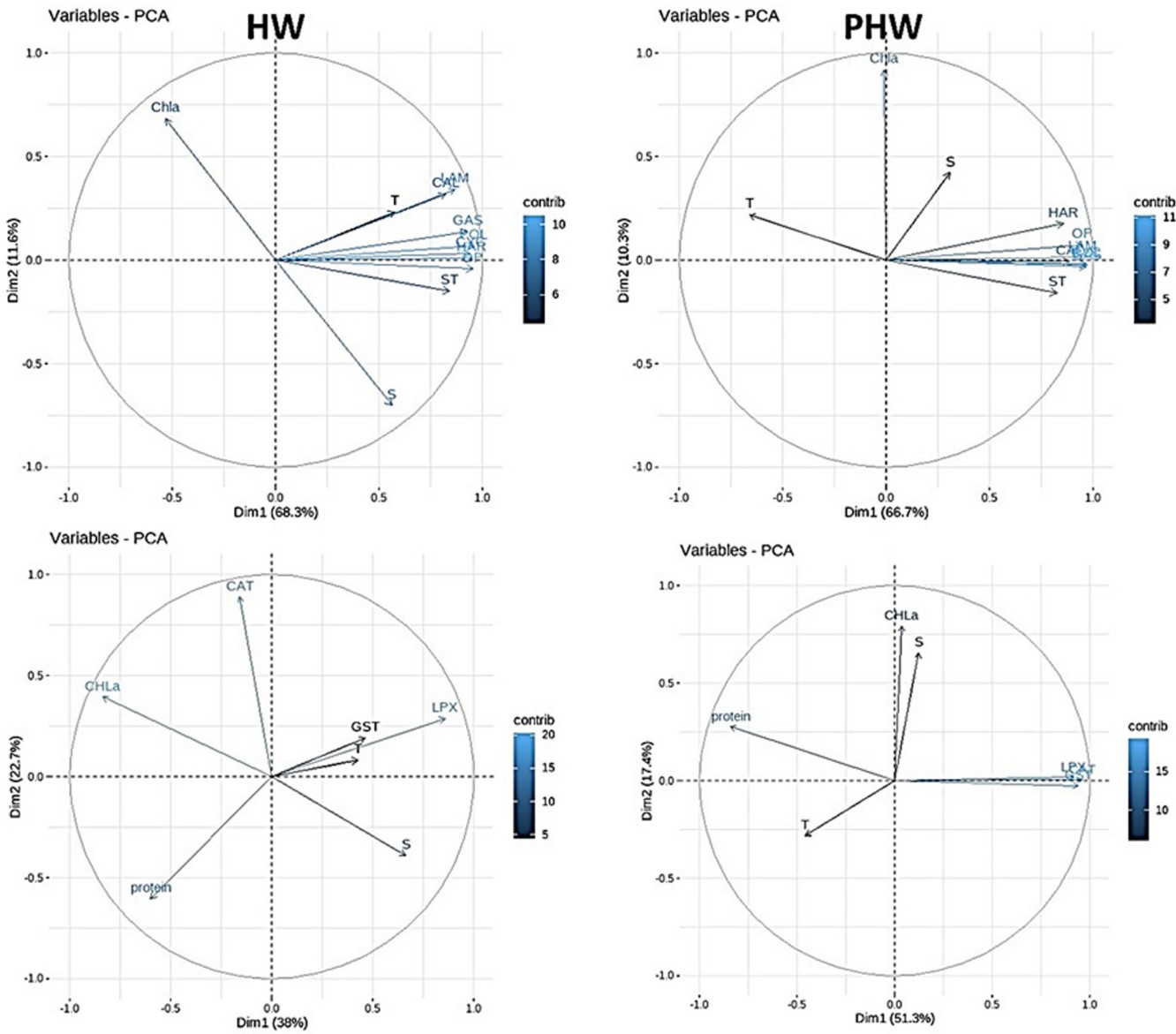

**Fig 7.** Principal Component Analyses (PCA) of the log transformed values of zootaxa (CAL = Calanoida, HAR = Harpacticoida, CYC = Cyclopoida, LAM = Lamellibranchia, GAS = Gastropoda, POL = Polychaeta), Sediment Traps (ST), Offspring Production (OP) and environmental-biological variables (T = temperature, S = salinity, Chla = chlorophyll *a* concentration) during the HW period (A) and the post-HW period (B), as well as biomarkers (GST = glutathione s-transferase, CAT = catalase, LPX = lipid peroxidation and proteins) and environmental-biological variables (T = temperature, S = salinity, Chla = chlorophyll *a* concentration), during the HW period (C) and the post-HW period (D). When variables are close to each other, they are positively correlated. When they are opposed, they are negatively correlated. When they are orthogonally located, they are not correlated. When variables are close to the center, they are not well represented by the analysis.

the heatwave period, when calanoid copepods were found in high numbers in the sediment traps and during the post-heatwave period when polychaete larvae dominated the water samples. The negative impact of higher seawater temperatures on calanoid copepods have already been reported. For example, a zooplankton community analysis in Lake Baikal indicated significant increases in water temperature and corresponding subtle declines in copepod abundances [56, 57]. Analysis of copepod time-series data from the eastern North Atlantic (Bay of Biscay and the Kattegat Sea) and the Mediterranean Sea (Saronikos Gulf) revealed a high temporal

**Table 3. Eigenvalues, percentage of variance and cumulative percentage of variance of Principal Component Analyses (PCA) of the log transformed values of zoo-taxa and biomarkers.**

| | | PCs | Eigenvalue | Percentage of variance | Cumulative percentage of variance |
|---|---|---|---|---|---|
| **BIOMARKERS** | **HW** | 1 | 2.661 | 38.009 | 38.009 |
| | | 2 | 1.591 | 22.725 | 60.734 |
| | | 3 | 0.989 | 14.121 | 74.855 |
| | | 4 | 0.902 | 12.888 | 87.744 |
| | | 5 | 0.435 | 6.208 | 93.952 |
| | | 6 | 0.268 | 3.824 | 97.777 |
| | | 7 | 0.156 | 2.223 | 100.000 |
| | **PHW** | 1 | 3.589 | 51.275 | 51.275 |
| | | 2 | 1.215 | 17.354 | 68.629 |
| | | 3 | 0.902 | 12.887 | 81.516 |
| | | 4 | 0.820 | 11.720 | 93.236 |
| | | 5 | 0.270 | 3.863 | 97.099 |
| | | 6 | 0.147 | 2.104 | 99.203 |
| | | 7 | 0.056 | 0.797 | 100.000 |
| **ZOOTAXA** | **HW** | 1 | 7.518 | 68.349 | 68.349 |
| | | 2 | 1.278 | 11.617 | 79.966 |
| | | 3 | 0.757 | 6.880 | 86.847 |
| | | 4 | 0.455 | 4.133 | 90.980 |
| | | 5 | 0.320 | 2.905 | 93.884 |
| | | 6 | 0.262 | 2.380 | 96.264 |
| | | 7 | 0.136 | 1.240 | 97.503 |
| | | 8 | 0.092 | 0.835 | 98.339 |
| | | 9 | 0.076 | 0.693 | 99.032 |
| | | 10 | 0.063 | 0.575 | 99.607 |
| | | 11 | 0.043 | 0.393 | 100.000 |
| | **PHW** | 1 | 7.332 | 66.654 | 66.654 |
| | | 2 | 1.130 | 10.270 | 76.925 |
| | | 3 | 0.898 | 8.165 | 85.090 |
| | | 4 | 0.642 | 5.832 | 90.922 |
| | | 5 | 0.333 | 3.023 | 93.945 |
| | | 6 | 0.263 | 2.390 | 96.335 |
| | | 7 | 0.216 | 1.960 | 98.295 |
| | | 8 | 0.068 | 0.618 | 98.913 |
| | | 9 | 0.058 | 0.525 | 99.438 |
| | | 10 | 0.032 | 0.289 | 99.728 |
| | | 11 | 0.030 | 0.272 | 100.000 |

turnover in the community composition and decreased similarity ('decay') over three decades [58]. Moreover, a studied time series (2009–2010, 2015–2016, 2018–2022) revealed a significant decrease (up to two-fold) in four of the dominant zooplankton species in the North Sea ecosystem, the calanoid copepods *Acartia clausi*, *Calanus helgolandicus*, *Centropages* spp., and *Temora longicornis*. However, another experimental study performed at the same location as the present study, where water temperature increase was tested without being followed by a recovery period, revealed that copepods dominated the zooplankton community, as well as a positive effect of warming on copepod development time was detected [26]. In addition, Lewandowska et al. [20] conducted an experiment for a period of one month, in which six mesocosms with

6˚C differences among them, showed that the copepod abundance increased in the warmer mesocosms. A small scale 22-day experiment with four different temperatures, found that copepod *Epischura lacustris* decreased with rising temperatures [59]. In the Gulf of Alaska, a two-year marine heatwave seemed to negatively affect the phytoplankton community, with neutral to positive effects on the zooplankton community [8, 60]. Thus, it seems that the responses to warming are species-specific and may be determined by whether the exact timing of warming coincides with critical life cycle stages or events. This suggests that an intimate knowledge of the life history of an organism is needed for an adequate explanation of population impacts and prediction of ecosystem responses [61].

The timing of heatwave events is a critical factor in determining their impact on zooplankton communities, with varying effects due to the distinct physiological states and life cycle stages of zooplankton during different seasons [62]. Our study has been focused mainly on spring/summer heatwaves, therefore it is crucial to consider the potential impacts of winter warming, which has been observed to occur more rapidly and can have severe effects on marine communities. For example, winter heatwaves might affect overwintering stages or disrupt the timing of life cycle events, leading to mismatches in predator-prey interactions and reproductive cycles [63]. Moreover, the combined effects of multiple stressors such as temperature changes, pollution, and nutrient fluctuations,can be more severe than the sum of their individual impacts mainly during the winter period [64].

Polychaete larval abundances in the present study increased considerably during the post heatwave period after the artificial heatwave terminated. Pansch et al. [65] showed that the abundance of a polychaete *Marenzelleria viridis* increased up to 50% during heatwave, suggesting the taxon to be highly tolerant to temperature rise, as well as rapid changes in temperature. There are several examples of polychaete species, so called marine ecosystem engineers that show high tolerance to different environmental changes [66–68]. Heatwaves, in general, favour suspension feeders, as well as predators with crawling or burrowing behaviour. Therefore, polychaetes seem tolerant to warming which partly could be due to their meroplanktonic life history [69]. In the Baltic Sea, where Pansch et al. [65] performed the study, *M. viridis* as an alien species, which further could positively affect its success during heatwaves. Kotta et al. [70] found a positive correlation between polychaete larval abundance and temperature in the Gulf of Riga of the Baltic Sea. To be noted is that in the current work, there was a statistically higher abundance of polychaete larvae also in the sediment traps during post-heatwave period, indicating that mortality was high also in the water column. Another reason for abundant numbers of polychaetes in sediment traps is diel vertical migration. Numerous studies have traditionally posited that polychaete larvae undergo passive dispersion in the water column owing to their constrained swimming capabilities. However, Abe et al. [71] have revealed that polychaete larvae exhibit a degree of control over their vertical distribution. This small-scale vertical migration is proposed to play a crucial role as a retention mechanism for polychaete larvae, diverging from the conventional understanding of their passive dispersal. Cole et al. [72] demonstrated that polychaetes benefitted from warming and stated that they seem to be successful also under ocean acidification (OA) regimes. Harpacticoid copepods were another successful group in the heatwave experiment, but knowledge about their heat tolerance is far less than that of Polychaeta. Although, they have been suggested as a valuable group for predicting climate changes [73], as there are not many studies examining on how the elevated temperature will impact the harpacticoid population, since most temperature experiments have been performed using the genus *Tisbe* or *Tigriopus* [74–77].

Thus, harpacticoid copepods and polychaete larvae may possess higher thermal tolerance thresholds, allowing them to maintain physiological functions at elevated temperatures [62]. These organisms might have more efficient stress response mechanisms, such as heat shock

proteins, that help protect cellular structures and functions during thermal stress [78]. More-over, they can exhibit behavioral adaptations, such as vertical migration to cooler water layers, which help them avoid the strongest effects of heatwaves [79]. More resilient species, like har-pacticoid copepods and polychaete larvae, may become more dominant, altering the balance of the ecosystem [80].

## Copepod grazing and offspring production

Copepod reproduction and growth are functions of the maternal dietary background and their feeding activity. Food quality and cell size are important components for reproductive effi-ciency and different responses may be related to dietary diversity [81, 82]. In this study, the potential copepod prey (diatoms, dinoflagellates and ciliates) changed during warming, which is in accordance with other studies performed in the mesocosms at the same site [26, 32]. Sou-lié et al. [32] reported that the heatwave seemed to stimulate zooplankton development and metabolism during heatwave [26, 83, 84], which resulted in higher zooplankton grazing during the post-heatwave. Grazing rates of *Acartia* on various prey found in this study fall within the range observed for this species in other oligotrophic environments [85–88] and a significant trend was observed from heatwave to post-heatwave on ciliates. Vidussi et al. [26] and Lewan-dowska et al. [20] found that ciliates declined as their potential predators appeared in meso-cosms. In our study, *A. clausi* strongly preferred ciliate prey; despite that they made up only 17% of the total prey community in the mesocosms. *Acartia* are selective foragers that switch between ambush feeding and filtration [89, 90]. Ciliates are microzooplankton that are pre-dicted to benefit from future warming, as they are small, part of the microbial loop, have fast growth rates and thrive in multiple environments [91–93]. Our study indicates an indirect effect of temperature-dependent mesozooplankton grazing implying that future heatwaves could alter the coupling between phytoplankton and their grazers, with major impacts on the structure of the microbial food web [94].

The impact of consecutive heat on zooplankton can be multifaceted and may vary based on the specific species, life stage, genotype, duration, and intensity of the heat exposure. Addition-ally, the ongoing and increased use and release of chemical contaminants in the environment present another significant threat [95]. Organisms exhibit diminished performance even before reaching their upper and lower critical limits [96]. Consequently, as temperatures devi-ate from the optimal range by increasing or decreasing, a gradual reduction in performance is observed [97]. Thus, it is expected that elevated temperature influences the rate of physiologi-cal processes and affects the growth and egg production of marine copepods [98, 99]. Rhyne et al. [100] found that temperature had a significant effect on nauplii production and survival. This is consistent with our results where offspring were most abundant during post-heatwave period, and confirms that egg hatching probably occurred faster during the heatwave, resulting in more offspring. On the other hand, Vidussi et al. [26] observed that the nauplii number decreased faster during warming, as the development time to the next stage was shorter. How-ever, Garzke et al. [101] observed that copepods' reproduction decreased with elevating tem-peratures. Siegle et al. [76] have also found that temperature *per se*, but not the duration of heat-wave exposure, negatively affected the reproduction of *Tigriopus californicus*. The feeding history of the initially enclosed copepod assemblages as well as the prey community within the mesocosms could possibly contribute to the results of our study compared to the aforemen-tioned studies. With regard of the prey community, Soulié et al. [32] found in another meso-cosm experiment with different experimental set-up that phytoplankton community structure was significantly affected during the post-heatwave period, as cyanobacteria seemed favoured in the heatwave treatment at the expense of dinoflagellates. This contrasts to what was

observed during the heatwave period when it was diatoms and prymnesiophytes that contributed to the increase in phytoplankton biomass. Regarding the copepod abundances, it is interesting to note the dominance of the egg-carrying copepods during the post-heatwave and the high mortality of the calanoids found in the sediment traps. Their occurrence suggests an adaptive advantage of egg-carriers versus broadcast spawners. Such an ecological success could have arisen if there was a reduction in the egg mortality in sac spawners as suggested by Kiørboe & Sabatini [102], or perhaps because the species possesses other advantages such as: relatively low metabolic demands (e.g., *Clausocalanus*, [103]); a wide spectrum of acceptable food types (e.g., *Oithona*, [104]); behavioral adaptations to avoid predation; a limited feeding specialization and relative longevity of the adults (e.g., *Oithona*, [105]).

## Zooplankton biomarkers

Temperature is a critical abiotic factor that has serious effects on the physiology of marine biota, as temperature can lead to high energy costs due to elevated respiration, which then again leads to higher ROS [106 and references therein]. All organisms have an optimal thermal window [107], and temperature affects all living organisms; when it gets warmer, the oxygen consumption, including respiration is enhanced (49% increase of community respiration in heatwave period [31]. Consequently, reactive oxygen species ROS that are harmful to the cell, can increase due to more intensive cellular metabolism, leading to oxidative stress [108]. Long-term stress can lead to mortality, which was observed during the heatwave period in the sediment trap (Fig 4).

As expected, lipid peroxidation (LPX) in the zooplankton community increased with time during the Heatwave [cf. 109–112]. The fact that LPX increased, but antioxidant glutathione-s-transferase GST did not, whereas catalase CAT did, is alarming, as it suggests that the cell did not have the full capacity to increase the antioxidant defence against elevating cellular ROS leading to direct cellular damage. Both GST and CAT did not rise during the Heatwave treatment, suggesting that the situation was stressful for the zooplankton community, and they were exposed to cell damage [33, 111, 112]. Our results suggest that the defence system starts to work again after the Heatwave period, whereas the community were not able to produce enough antioxidants (CAT and GST) as LPX still was high. Future work should look into the degree of elevated temperature that causes damage to lipids, as heatwaves will become stronger and more intensive in the future [1]. In the current experiment we used ∼3 degrees increase, whereas 5 degrees could be even more detrimental.

Salinity increased approximately ½ unit during the 20-d experiment in all mesocosms. In the PCA, we show that salinity had quite a strong effect on the different biomarker enzymes. There are some studies looking into salinity effects on oxidative stress enzymes in marine biota. The Baltic clam *Macoma balthica* shows increased catalase CAT during decreasing salinity [113], whereas von Weissenberg et al. [112] detected combined effects of salinity and sampling date (i.e., temperature) on copepod reproductive rates. Yang et al. [114] found an interaction between salinity and temperature on both CAT and superoxide dismutase (SOD) in pearl oysters. Even small salinity changes can obviously cause a response in the defence system [113 and current work].

## Conclusions

In conclusion, the results of this study provide valuable insights into the effects of elevated temperature (heatwave) on zooplankton communities, their offspring production, and the associated biomarkers in a mesocosm experiment. The composition of zooplankton communities underwent significant changes over time. Copepods, polychaetes and gastropods played

a dominant role in the communities, and their abundances were influenced by the heatwave and even in post-heatwave conditions. Zooplankton abundance in the sediment traps increased over time, indicating high mortality under the heatwave. Offspring production, including copepod eggs and nauplii, exhibited significant variations across the control, heatwave, and post-heatwave conditions. Higher numbers of offspring were observed during the heatwave and production was even higher during the post-heatwave period. The grazing behavior of copepods was influenced by prey type, with higher clearance rates observed on ciliates and dinoflagellates. Biomarker analysis revealed changes in lipid peroxidation (LPX) levels in zooplankton communities. LPX levels were high during the heatwave but started to decrease during the post-heatwave. Catalase (CAT) enzyme levels also exhibited variations, with a significant increase during the post-heatwave period.

Overall, these findings suggest that elevated temperature experienced during a heatwave, even though being short-term and of moderate intensity, can significantly impact zooplankton communities, their reproductive success and the biochemical responses of these organisms. It's essential to note that the specific responses of zooplankton to consecutive heat events can vary among different species and environmental contexts. Additionally, the potential long-term consequences on ecosystems and food webs need to be considered when evaluating the effects of sustained warming on zooplankton populations. These insights are valuable in understanding the potential consequences of temperature variations on aquatic ecosystems, notably in the context of global warming, and highlight the need for continued research in the Mediterranean Sea. Understanding the effects of heatwaves on zooplankton communities requires a comprehensive approach that considers seasonal changes, species-specific adaptations, and the interaction with multiple anthropogenic stressors. Future research should focus on long-term monitoring and experiments across different seasons to capture the full range of responses and inform effective management and conservation strategies for maintaining marine ecosystem health in the face of climate change.

## Acknowledgments

We would like to thank Dr. Sébastien Mas, David Parin, Rémi Valdès, Solenn Soriano from MEDIMEER and Dr. Tanguy Soulié for valuable help with the mesocosm setup, laboratory analyses, sampling, and sensor installment prior to the experiment. Thanks to Tanguy Soulié and Inés Ricard for HPLC pigment analysis and to Luigia Verde for pigment sampling. In addition, special thanks to Sébastien Mas also for all help with practicalities during the stay of guest scientists at the Sète Marine Station of OSU OREME (CNRS, Univ. Montpellier, IRD, IRSTEA).

## Author Contributions

**Conceptualization:** Soultana Zervoudaki, Behzad Mostajir, Francesca Vidussi, Jonna Engström-Öst.

**Data curation:** Soultana Zervoudaki, Maria Protopapa, Andriana Koutsandrea, Anna Jansson, Ella von Weissenberg, Georgios Fyttis, Athanasia Sakavara, Charitomeni Chariati, Jonna Engström-Öst.

**Formal analysis:** Soultana Zervoudaki, Maria Protopapa, Andriana Koutsandrea, Ella von Weissenberg, Katja Anttila, Pauline Bourdin, Jonna Engström-Öst.

**Funding acquisition:** Soultana Zervoudaki, Maria Protopapa, Anna Jansson, Ella von Weissenberg, Georgios Fyttis, Athanasia Sakavara, Behzad Mostajir, Francesca Vidussi, Jonna Engström-Öst.

**Investigation:** Soultana Zervoudaki, Jonna Engström-Öst.

**Methodology:** Soultana Zervoudaki, Maria Protopapa, Andriana Koutsandrea, Anna Jansson, Ella von Weissenberg, Georgios Fyttis, Athanasia Sakavara, Kostas Kavakakis, Behzad Mostajir, Francesca Vidussi, Jonna Engström-Öst.

**Project administration:** Behzad Mostajir, Francesca Vidussi.

**Supervision:** Soultana Zervoudaki, Behzad Mostajir, Francesca Vidussi.

**Validation:** Soultana Zervoudaki.

**Visualization:** Soultana Zervoudaki, Maria Protopapa, Jonna Engström-Öst.

**Writing – original draft:** Soultana Zervoudaki, Jonna Engström-Öst.

**Writing – review & editing:** Soultana Zervoudaki, Maria Protopapa, Andriana Koutsandrea, Anna Jansson, Ella von Weissenberg, Georgios Fyttis, Athanasia Sakavara, Kostas Kavakakis, Charitomeni Chariati, Katja Anttila, Pauline Bourdin, Behzad Mostajir, Francesca Vidussi, Jonna Engström-Öst.

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
