## [Decision Letter · Decision Letter 0]

14 May 2024

PONE-D-24-05200Zooplankton responses to simulated marine heatwave in the Mediterranean Sea using in situ mesocosmsPLOS ONE

Dear Dr. ZERVOUDAKI,

Thank you for submitting your manuscript to PLOS ONE. After careful consideration, we feel that it has merit but does not fully meet PLOS ONE’s publication criteria as it currently stands. Therefore, we invite you to submit a revised version of the manuscript that addresses the points raised during the review process.

I apologize for the delay in reaching a decision on your manuscript.  I was able to secure only one review. The reviewer raises several issues. First, please ensure that your methods are sufficiently clear that the reader does not need to consult other papers to understand your methods. Give enough detail for the reader to be able to replicate your experiments. The reviewer also raises issues regarding the statistical treatment of the data, and the rationale for the experiment conditions. Finally, the reviewer recommends consulting recent work on marine heatwaves that may help you better frame the implications of your study. I urge you to address the reviewer's concerns and suggestions.

We look forward to receiving your revised manuscript.

Kind regards,

Hans G. Dam, Ph. D.

Academic Editor

PLOS ONE

Journal Requirements:

2. Thank you for stating the following financial disclosure: "The work was funded by the AQUACOSM-plus project, which has received funding from the European Union’s Horizon 2020 Research and Innovation Program (H2020/2017–2020) under grant agreement nr. 731065"  

3. Thank you for stating the following in the Acknowledgments Section of your manuscript: "The work was funded by the AQUACOSM-plus project, which has received funding from the European Union’s Horizon 2020 Research and Innovation Program (H2020/2017–2020) under grant agreement nr. 731065"  

Please remove any funding-related text from the manuscript and let us know how you would like to update your Funding Statement. Currently, your Funding Statement reads as follows: "The work was funded by the AQUACOSM-plus project, which has received funding from the European Union’s Horizon 2020 Research and Innovation Program (H2020/2017–2020) under grant agreement nr. 731065"  

4. Please note that your Data Availability Statement is currently missing [the repository name and/or the DOI/accession number of each dataset OR a direct link to access each database]. If your manuscript is accepted for publication, you will be asked to provide these details on a very short timeline. We therefore suggest that you provide this information now, though we will not hold up the peer review process if you are unable.

Reviewers' comments:

Reviewer's Responses to Questions

**Comments to the Author**

1. Is the manuscript technically sound, and do the data support the conclusions?

Reviewer #1: Partly

2. Has the statistical analysis been performed appropriately and rigorously? 

Reviewer #1: Yes

3. Have the authors made all data underlying the findings in their manuscript fully available?

Reviewer #1: No

4. Is the manuscript presented in an intelligible fashion and written in standard English?

Reviewer #1: Yes

5. Review Comments to the Author

Reviewer #1: This study investigated the ecological and physiological impacts of a simulated marine heatwave on the zooplankton community in the Thau Lagoon. I find it interesting and have a few suggestions for the revision.

L86-88: This should be better formulated as hypotheses and also more specific about the direction of the effects. For example, what would you expect in the changes of the community composition under a marine heatwave?

Methods

It is quite hard to understand the setup of the experiment. Authors referred to three published papers, but it is necessary to provide brief descriptions, at least to the point that the reader can understand how the experiment was set up, and whether or not there would be any confounding effects or even pseudo-replications.

There are also no justifications for choosing 10 days of a simulated marine heatwave and 10 days post marine heatwave period?

I was wondering why the authors decided to discard the results > 10% of the mean coefficient variation percentage (CV%). How many data were discarded and would there be any extreme values that could have changed the overall statistical results?

It doesn’t sound right to me that the salinity was included as a fixed effect, but more like a co-variate. Also Figure 1b, please make the y-axis from 0. It looks like the salinity increased a lot over time, but it was minor.

Results and discussions

L264-266: Do you mean the polychaeta, gastropoda and bivalvia larvae?

What are the physiological mechanisms underlying the high mortality of zooplankton during the heatwave period while the performance of harpacticoid copepods and polychaete larvae was better? What would be the ecological consequences of this? The heat stress can substantially shorten the lifespan of calanoid copepods, up to 50% (Truong, K. N., Vu, N.-A., Doan, N. X., Le, M.-H., Vu, M. T. T., & Dinh, K. V. (2020). Predator cues increase negative effects of a simulated marine heatwave on tropical zooplankton. Journal of Experimental Marine Biology and Ecology, 530-531, 151415. doi:https://doi.org/10.1016/j.jembe.2020.151415).

L533-534: I agree (see e.g., Dinh, K. V., Konestabo, H. S., Borgå, K., Hylland, K., Macaulay, S. J., Jackson, M. C., Verheyen, J., & Stoks, R. (2022). Interactive effects of warming and pollutants on marine and freshwater invertebrates. Current Pollution Reports, 8, 341-359. doi:10.1007/s40726-022-00245-4 ).

It is a bit hard to follow the main results and discussion about the direct vs delayed effects of the simulated marine heatwave. I would strongly suggest authors revise to make it clear, particularly any potential recovery from a heatwave event.

The recommendation for the need for continuing research in the Mediterranean Sea is a bit weak without clear suggestions on what exactly is needed and how to conduct future research in the best way possible.

Beyond this, I would like to discuss a bit more about timing of the heatwave effect. The experiment was conducted in the summer time, while previous studies have shown that winter warming is faster, and could disrupt coastal marine fish community structure (see e.g., Clark, N. J., Kerry, J. T., & Fraser, C. I. (2020). Rapid winter warming could disrupt coastal marine fish community structure. Nature Climate Change, 10(9), 862-867. doi:10.1038/s41558-020-0838-5). This could be even worse when combining with other anthropogenic stressors (Dinh, K. V., Albini, D., Orr, J. A., Macaulay, S. J., Rillig, M. C., Borgå, K., & Jackson, M. C. (2023). Winter is coming: Interactions of multiple stressors in winter and implications for the natural world. Global Change Biology, 29(24), 6834-6845. doi:https://doi.org/10.1111/gcb.16956 ). It is important to discuss the heatwave effects on zooplankton community, considering the seasonal changes and adaptations (Sasaki, M. C., M. Finiguerra, & H. G. Dam. Seasonally variable thermal performance curves prevent adverse effects of heatwaves. https://www.biorxiv.org/content/10.1101/2023.05.09.540050v1.abstract ) in the multiple stressor context.

6. PLOS authors have the option to publish the peer review history of their article (what does this mean?). If published, this will include your full peer review and any attached files.

Reviewer #1: No

---

## [Author Response · Author response to Decision Letter 0]

26 Jul 2024

Dear Editor,

Please find below a point-by-point response to the Reviewer’s comments. In very few cases where we did not follow a recommendation, a detailed rebuttal is given.

Our replies to the Reviewers’ comments are highlighted in blue text. All amendments have been inserted throughout the revised manuscript in red colour indicting the new lines number. The amendments can be found in the different sections of the manuscript according to the reviewer’s comments.

Reviewer: This study investigated the ecological and physiological impacts of a simulated marine heatwave on the zooplankton community in the Thau Lagoon. I find it interesting and have a few suggestions for the revision.

We thank the reviewer that our research on ecological and physiological impacts of a simulated marine heatwave on the zooplankton community is interesting and can contribute to new knowledge. We followed the suggested comments in the revised version of the manuscript.

L86-88: This should be better formulated as hypotheses and also more specific about the direction of the effects. For example, what would you expect in the changes of the community composition under a marine heatwave?

REPLY: Thank you for the comment. We deleted the sentences from the line 91 to 100, we added the following text: “Heatwaves can cause significant stress on zooplankton organisms and can substantially shorten the lifespan of calanoid copepods by up to 50% (Truong et al. 2020). This might be due to increased metabolic demand as higher temperatures accelerate metabolic processes, leading to faster energy depletion and reduced lifespan (Pörtner et al., 2017). Also, elevated temperatures can impair reproductive processes, reducing the number of viable offspring and overall population sustainability (Almeda et al., 2014) and prolonged exposure to heat stress can cause cumulative physiological damage, leading to premature death (Hochachka & Somero, 2002). This can lead to shifts in community structure and thus changes in carbon sequestration and disruption of the food webs”

And we reformulated the aim of our study: In this study we investigated the zooplankton community composition, vital rates (including grazing, production, survival, and oxidative stress), and the quantity and quality of zooplankton prey, assessing whether these factors were impacted by the heatwave.

Methods

It is quite hard to understand the setup of the experiment. Authors referred to three published papers, but it is necessary to provide brief descriptions, at least to the point that the reader can understand how the experiment was set up, and whether or not there would be any confounding effects or even pseudo-replications.

REPLY: We thank you for the comment, we reformulated the methodology for the description of the mesocosm set up. 

Lines 108-111: The mesocosm field experiment studying heatwave effects on the plankton community was conducted using the MEDIMEER platform (Mediterranean platform for Marine Ecosystems Experimental Research) based at the marine station of Sète (SMEL, University of Montpellier 2, 43° 24’ 49” N, 3° 41’ 19” E) in summer 2019. 

Lines 120-121: Technical details of the platform and information on data monitoring of infrastructure are reported by Nouguier et al. [34], Vidussi et al. [24], and Mostajir et al. [25].

Lines 129-136: Two treatments, each in triplicate, were applied to six mesocosms. During the first 10 d of the experiment (d1–d10), the water temperature was raised +3°C in three mesocosms (heatwave H), while the three remaining mesocosms had natural lagoon temperature (control C). After this 10 d period (d11–d20), the temperature was returned to ambient water temperature (post-heatwave PH). During this period, the “control mesocosms” experienced the in situ lagoon temperature. In addition, two incubation mesocosms with the same water temperature as Control and Heatwave experimental mesocosms were established to incubate several samples, thereby avoiding contamination of the experimental mesocosms.

Lines 136-139: Detailed information on the installed high-frequency sensors for measuring temperature, chlorophyll a, conductivity, and light, as well as results on these data and nutrients are found in Soulié et al. [29]. In brief,

There are also no justifications for choosing 10 days of a simulated marine heatwave and 10 days post marine heatwave period?

REPLY: Thank you for this question; few studies have looked into the processes, such as recovery, occurring after a marine heatwave. We selected ten days as the length of the heatwave, as heatwaves are forecasted to get longer in the future (Soulié et al. 2022). Ten days of post-heatwave was also suitable, considering the total length of the mesocosm experiment.

I was wondering why the authors decided to discard the results > 10% of the mean coefficient variation percentage (CV%). How much data were discarded and would there be any extreme values that could have changed the overall statistical results?

REPLY: Thanks for asking! The 10% discard concerns here only the biomarker data (oxidative stress or antioxidant responses); the reason for some discard is not extreme values, instead it is related to the amount of protein in the sample that conveys information about the biomass. If protein is low or under detection, it does not give a reliable value of the bioassay (could mislead), and the measurement has to be discarded.

It doesn’t sound right to me that the salinity was included as a fixed effect, but more like a co-variate. Also Figure 1b, please make the y-axis from 0. It looks like the salinity increased a lot over time, but it was minor.

REPLY: Thank you for the comment. In this case we are interested in both the treatment (heatwave) and salinity. They are called fixed effects, but salinity functions here as a covariate and in the linear mixed model set-up, the “fixed effect" is synonymous to "co-variate". 

Concerning the scale of the salinity figure; we agree that it would be optimal to scale the salinity from 0 to 40. Unfortunately, that will not work as the data points would be clustered and hard to read, and as the changes in salinity occur between 38.2 and 38.6. We reformulated the title of the legend for the figure 6 and we added a note to the figure legend that scale on the x-axis should be noted “Fig. 6. Biomarker enzymes measured from >200 um plankton community during heatwave experiment in Control C and Treatment T, consisting of 10 days heatwave following 10 days of post-heatwave. A) Lipid peroxidation (LPX), FOX mg protein-1 during Period 1 (Heatwave) and Period 2 (Post-heatwave), and B) LPX as a relationship with salinity, C) catalase (CAT) μmol min-1 mg-1 , as well as D) protein mg-1 against increasing salinity. FOX = ferrous oxidation - xylenol orange. Note salinity scale in figs. B and D. will add .  [line 411]

Results and discussions

L264-266: Do you mean the polychaeta, gastropoda and bivalvia larvae? 

Reply: Yes we have changed lines 264-266 accordingly : At the start of the experiment (D0), total zooplankton abundance in the Thau lagoon was 1227 ± 358 ind. m-3 and the community was dominated by Copepoda, Cladocera, Polychaeta larvae, Gastropoda larvae and Bivalvia larvae (Lines 281-282)

What are the physiological mechanisms underlying the high mortality of zooplankton during the heatwave period while the performance of harpacticoid copepods and polychaete larvae was better? What would be the ecological consequences of this? The heat stress can substantially shorten the lifespan of calanoid copepods, up to 50% (Truong, K. N., Vu, N.-A., Doan, N. X., Le, M.-H., Vu, M. T. T., & Dinh, K. V. (2020). Predator cues increase negative effects of a simulated marine heatwave on tropical zooplankton. Journal of Experimental Marine Biology and Ecology, 530-531, 151415. doi:https://doi.org/10.1016/j.jembe.2020.151415). 

REPLY: Heatwaves, in general, favour suspension feeders, as well as predators with a crawling or burrowing behaviour. Therefore, polychaetes seem tolerant to warming which partly could be due to their meroplanktonic life history. According to the bibliography we have added the following text:

Lines 546-553: Thus, harpacticoid copepods and polychaete larvae may possess higher thermal tolerance thresholds, allowing them to maintain physiological functions at elevated temperatures (Sasaki et al., 2023). These organisms might have more efficient stress response mechanisms, such as heat shock proteins, that help protect cellular structures and functions during thermal stress (Tomanek, 2010). Moreover, they can exhibit behavioral adaptations, such as vertical migration to cooler water layers, which help them avoid the strongest effects of heatwaves (Pearre, 2003). More resilient species, like harpacticoid copepods and polychaete larvae, may become more dominant, altering the balance of the ecosystem (Sommer et al., 2017). 

L533-534: I agree (see e.g., Dinh, K. V., Konestabo, H. S., Borgå, K., Hylland, K., Macaulay, S. J., Jackson, M. C., Verheyen, J., & Stoks, R. (2022). Interactive effects of warming and pollutants on marine and freshwater invertebrates. Current Pollution Reports, 8, 341-359. doi:10.1007/s40726-022-00245-4 ).

REPLY: Thanks for this valuable reference. We included it in the manuscript and rephrased the sentence accordingly: (Lines 574-577) “The impact of consecutive heat on zooplankton can be multifaceted and may vary based on the specific species, life stage, genotype, duration, and intensity of the heat exposure. Additionally, the ongoing and increased use and release of chemical contaminants in the environment present another significant threat (Dihn et al 2022).

It is a bit hard to follow the main results and discussion about the direct vs delayed effects of the simulated marine heatwave. I would strongly suggest authors revise to make it clear, particularly any potential recovery from a heatwave event.

REPLY: Thanks for the comment. We decided to remove the word recovery from the manuscript as that was not exactly what we studied. Recovery was one of the main aims of the Aquacosm Summer-in-spring 2019 experiment (Soulié, T., Vidussi, F., Mas, S., & Mostajir, B. (2022). Functional stability of a coastal Mediterranean plankton community during an experimental marine heatwave. Frontiers in Marine Science, 9, 831496), but not part of our paper. 

The recommendation for the need for continuing research in the Mediterranean Sea is a bit weak without clear suggestions on what exactly is needed and how to conduct future research in the best way possible. 

REPLY: Thank you for this comment. We included the following text in the introduction (Lines 64-72): Studying the relationship between plankton and marine heatwaves in the Mediterranean Sea is critical due to the foundational role plankton play in marine ecosystems and their sensitivity to temperature changes. Changes in plankton dynamics can affect biogeochemical cycles, including carbon sequestration, thus influencing broader climate regulation processes (Hinder et al., 2012). The Mediterranean Sea's semi-enclosed nature and high biodiversity make it a crucial region for understanding these impacts. Researching plankton responses to marine heatwaves in this area can provide insights into the resilience and adaptability of marine ecosystems under climate change pressures and inform about conservation and management strategies (Basterretxea et al., 2019). 

Hinder, S. L., et al. (2012). "Changes in marine dinoflagellate and diatom abundance under climate change." Nature Climate Change.

Basterretxea, G., et al. (2019). Plankton response to warming in the NW Mediterranean Sea. Marine Ecology Progress Series.

Beyond this, I would like to discuss a bit more about timing of the heatwave effect. The experiment was conducted in the summer time, while previous studies have shown that winter warming is faster and could disrupt coastal marine fish community structure (see e.g., Clark, N. J., Kerry, J. T., & Fraser, C. I. (2020). Rapid winter warming could disrupt coastal marine fish community structure. Nature Climate Change, 10(9), 862-867. doi:10.1038/s41558-020-0838-5). This could be even worse when combining with other anthropogenic stressors (Dinh, K. V., Albini, D., Orr, J. A., Macaulay, S. J., Rillig, M. C., Borgå, K., & Jackson, M. C. (2023). Winter is coming: Interactions of mult iple stressors in winter and implications for the natural world. Global Change Biology, 29(24), 6834-6845. doi:https://doi.org/10.1111/gcb.16956 ). It is important to discuss the heatwave effects on zooplankton community, considering the seasonal changes and adaptations (Sasaki, M. C., M. Finiguerra, & H. G. Dam. Seasonally variable thermal performance curves prevent adverse effects of heatwaves. https://www.biorxiv.org/content/10.1101/2023.05.09.540050v1.abstract ) in the multiple stressor context.

REPLY: Thank you for the comment. We have included a discussion on the suggested subject.

Line 511-520: In addition, the timing of heatwave events is a critical factor in determining their impact on zooplankton communities, with varying effects due to the distinct physiological states and life cycle stages of zooplankton during different seasons (Sasaki et al., 2023). Our study has been focused mainly on spring/summer heatwaves, therefore it is crucial to consider the potential impacts of winter warming, which has been observed to occur more rapidly and can have severe effects on marine communities. For example, winter heatwaves might affect overwintering stages or disrupt the timing of life cycle events, leading to mismatches in predator-prey interactions and reproductive cycles (Clark et al., 2020). Moreover, the combined effects of multiple stressors such as temperature changes, pollution, and nutrient fluctuations,can be more severe than the sum of their individual impacts mainly during the winter period (Dinh et al., 2023).

In addition, we have modified the conclusion accordingly including the following sentences: Lines 657-662: Understanding the effects of heatwaves on zooplankton communities requires a comprehensive approach that considers seasonal changes, species-specific adaptations, and the interaction with multiple anthropogenic stressors. Future research should focus on long-term monitoring and experiments across different seasons to capture the full range of responses and inform effective management and conservation strategies for maintaining marine ecosystem health in the face of climate change.

---

## [Editor Report · Decision Letter 1]

1 Aug 2024

Zooplankton responses to simulated marine heatwave in the Mediterranean Sea using in situ mesocosms

PONE-D-24-05200R1

Dear Dr. ZERVOUDAKI,

We’re pleased to inform you that your manuscript has been judged scientifically suitable for publication and will be formally accepted for publication once it meets all outstanding technical requirements.

Kind regards,

Hans G. Dam, Ph. D.

Academic Editor

PLOS ONE
---

## [Editor Report · Acceptance letter]

15 Aug 2024

PONE-D-24-05200R1 

PLOS ONE

Dear Dr. Zervoudaki, 

I'm pleased to inform you that your manuscript has been deemed suitable for publication in PLOS ONE. Congratulations! Your manuscript is now being handed over to our production team.

Kind regards, 

on behalf of

Dr. Hans G. Dam 

Academic Editor

PLOS ONE